# Trauma, Post-Traumatic Stress Disorder, and Mental Health Care of Asylum Seekers

**DOI:** 10.3390/ijerph182010661

**Published:** 2021-10-12

**Authors:** Rafael Youngmann, Rachel Bachner-Melman, Lilac Lev-Ari, Hadar Tzur, Ravit Hileli, Ido Lurie

**Affiliations:** 1Clinical Psychology Graduate Program, Ruppin Academic Center, Emek-Hefer 4015000, Israel; rachel.bachner@mail.huji.ac.il (R.B.-M.); ldlevari@gmail.com (L.L.-A.); hadartzur@gmail.com (H.T.); 2School of Social Work and Social Welfare, Hebrew University of Jerusalem, Mount Scopus, Jerusalem 91905, Israel; 3Beer Ya’akov-Nez Tziona Mental Health Center, Beer Yaakov 70350, Israel; ravitkruvit@gmail.com; 4Department of Psychiatry, Sackler School of Medicine, Tel Aviv University, Tel Aviv 6812509, Israel; ido.lurie@gmail.com; 5Shalvata Mental Health Center, Hod Hasharon 69978, Israel

**Keywords:** asylum seekers, Eritrea, Sudan, PTSD, trauma, mental health care

## Abstract

Asylum seekers in Israel from East Africa frequently experienced traumatic events along their journey, particularly in the Sinai Peninsula, where they were subjected to trafficking and torture. Exposure to trauma has implications for rights that are contingent on refugee status. This retrospective chart review aimed to characterize the types of traumas experienced by 219 asylum seekers (149 men) from Eritrea and Sudan who sought treatment at a specialized mental health clinic in Israel, and to compare the mental health of trauma victims (*n* = 168) with that of non-trauma victims (*n* = 53). About 76.7% of the asylum seekers had experienced at least one traumatic event, of whom 56.5% were diagnosed with post-traumatic stress disorder (PTSD). Most reported traumas were experienced en route in the Sinai, rather than in the country of origin or Israel. Few clinical differences were observed between trauma victims and non-trauma victims, or between trauma victims with and without a PTSD diagnosis. Our findings emphasize the importance of accessibility to mental and other health services for asylum seekers. Governmental policies and international conventions on the definition of human trafficking may need to be revised, as well as asylum seekers’ rights and access to health services related to visa status.

## 1. Introduction

The number of international migrants worldwide is growing, and currently exceeds 244 million [1]. Refugees and asylum seekers constitute one of the largest migratory movements, driven by violence, insecurity, and armed conflict [2]. Refugees are people who left their country of origin to avoid life-threatening circumstances or persecution for political, religious, or other reasons, and are recognized under the 1951 Convention on the Status of Refugees [3]. Asylum seekers have applied for but have not yet been granted refugee status. In 2020, there were 82.4 million refugees and 4.1 million asylum seekers worldwide [4].

As part of the global trend of migration, and in the wake of war and genocide in Darfur, and the political situation in Eritrea, the number of asylum seekers arriving in Israel from East Africa has increased significantly since 2006 [5,6]. Between 2006 and 2016, approximately 64,318 asylum seekers entered Israel, of whom 40,274 still resided there in 2016 [7]; 72% (29,014) from Eritrea, 20% (8002) from Sudan, 0.3% (121) from Ethiopia, 5.8% (2349) from other African countries and 2% (556) from elsewhere [7]. Based on the principle of non-refoulement, 91% of asylum seekers from Sudan and Eritrea have been granted group protection, including the right to remain in Israel until their home countries are deemed safe. The Israeli government has, however, adopted restrictive policies towards them, excluding them from full participation in Israel’s social, political and health systems, with financial and psychological implications [6,8].

An estimated 7000 of the 52,961 asylum seekers who arrived in Israel between 2009 and 2013 [9] were exposed to kidnapping, torture, and human trafficking during their journey through the Sinai desert [6]. An estimated 4000 did not survive the journey [10]. Asylum seekers exploited in this way are defined by the UN conventions as “Smuggling Victims”, “Torture Victims”, or “Human Trafficking Victims” [11]. Only those recognized as “Trafficking Victims” or “Victims of Holding in Conditions of Slavery” by the State of Israel are legally entitled to shelter and rehabilitation services, including health insurance and a valid work permit, for one year [12], regardless of their physical and mental health status [13]. According to the UN Convention on the rights of refugees to health care [3], ratified by Israel in 1958 [14], asylum seekers who were victims of torture or human trafficking are entitled to recognition as refugees with rights to medical care and other services.

Asylum seekers living in Israel have very slim chances of receiving refugee status. In 2016, only 500 of the 37,016 asylum seekers from Sudan and Eritrea were officially recognized as “Trafficking Victims” or “Victims of Holding in Conditions of Slavery” [15]. These asylum seekers included 5000–7000 victims of torture, human trafficking and kidnapping in Sinai or their country of origin. No asylum seekers, however, were officially recognized as victims of human torture [15]. Of 13,764 asylum requests submitted by people from Sudan and Eritrea by July 2017, only ten had refugee status in 2018 [16], and therefore access to medical services. Others were granted access to health services in medical or psychiatric emergencies only [17]. Identifying victims of trafficking remains a major challenge due to the ambiguity of the international definition of exploitation [18], difficulties in defining types of victims in the Sinai desert [19,20], victims’ lack of awareness of their rights [13], and restrictions imposed by the Israeli government on asylum seekers seeking recognition as trafficking victims [21].

The process of migration involves significant socio-psychological pressures that affect migrants’ mental state. Asylum seekers and trafficking victims are particularly vulnerable to trauma-related disorders including post-traumatic stress disorder (PTSD), depression, somatic disorders, eating disorders, substance abuse (such as self-medication), self-injurious behavior, suicidality, and psychotic disorders [22,23,24,25,26,27,28,29,30,31]. As elsewhere, asylum seekers in Israel have high rates of emotional distress and psychiatric disorders [32,33]. Victims of torture experienced trauma directly or vicariously [6]. In addition, as a visible minority, African asylum seekers in Israel are vulnerable to racism and rejection by the local population [34,35]. Asylum seekers undoubtedly need social support, employment, and medical treatment to cope with post-migration challenges [36,37].

Most asylum seekers in Israel are not eligible for national health insurance, which provides psychiatric treatment [36,38]. One of their main options for psychiatric care is *Gesher* (“Bridge” in Hebrew), an adult (>18 years) psychiatric clinic established by the Ministry of Health in 2014 in Tel Aviv-Jaffa, for asylum seekers and other undocumented migrants [39]. The goal of the clinic is to deliver culturally competent and trauma-sensitive mental health treatment that includes the work of cultural brokers. A description of the clinic and its activities can be found elsewhere [40].

This study aimed to characterize asylum seekers mainly from Eritrea and Sudan who arrived in Israel through the Sinai Peninsula, between 2007 and 2013, including the types of traumas they experienced and the mental health care they received. These aims are of particular importance in Israel, where asylum seekers, referred to by the Ministry of the Interior as “infiltrators”, have almost no access to mental health services.

We hypothesized that compared to asylum seekers who were not trauma victims, trauma victims would:Be at greater risk for PTSD.Have more consultations.Receive psychiatric medication more frequently.Report more drug/alcohol use.Have more suicidal ideation.Have more psychiatric hospitalizations.Emotional support (operationalized by having a partner in Israel or by being employed) would be associated with a reduction in the risk for a PTSD diagnosis, number of clinic consultations, receiving psychiatric medications, drug and alcohol use, suicidal ideation, and psychiatric hospitalizations in Israel.

These outcomes have important implications for promoting the initiative to allow asylum seekers in Israel and elsewhere to benefit from health insurance and health services.

## 2. Materials and Methods

### 2.1. Measures

Demographic data were taken from patients’ files: country of origin, age, gender, years of education, marital status, partner in Israel (yes/no/missing), full or partial employment, and religion (Muslim/Christian/other).

Categories of victims were defined as follows, based on international definitions [41] and the law in the State of Israel [8]:Trafficking victims had experienced recruitment, transportation, transfer, harboring, or receipt of persons using threat or force, coercion, abduction, fraud, deception or abuse of power, for exploitation [41].Torture victims had experienced repeated physical and mental violence, and suffered severe pain.Trafficking and torture victims had experienced both trafficking and torture.Sexual violence and torture victims had experienced both sexual violence and torture, involving severe physical or mental pain inflicted by others, from rape, inhuman, or degrading treatment [10].Smuggling victims had paid Bedouins in the Sinai Peninsula to be smuggled into Israel.Persecution victims had been imprisoned, almost killed, lost consciousness or forced to separate from family [42].Military trauma victims had experienced traumatic combat events in their country of origin.Civil trauma victims had experienced trauma of a criminal nature or an accident (work, car, etc.)

Place of trauma: country of origin; en route to Israel (Sinai); Israel.

PTSD was diagnosed by a mental health professional according to ICD-10 criteria (F43.1.) [43].

Number of consultations was the number of appointments with a mental health professional at the *Gesher* Clinic.

Additional variables: Psychiatric medications (yes/no); current drug abuse (yes/no); current alcohol use (yes/no); suicidal ideation (yes/no); and one or more psychiatric inpatient admissions in Israel (yes/no).

### 2.2. Procedure

The study was approved by the Internal Review Boards of the Abarbanel Mental Health Center and the Ruppin Academic Center [TASHAZ 26]. The data was collected and coded from the files by H.T. and R.H. Anonymity was strictly respected. Inter-rater reliability for variable coding between the coders for a random sample of 10 files, evaluated by H.T and I.L was 100%.

### 2.3. Data Analysis

Group comparisons were reported by the percentages within each group. Chi Square tests were conducted to compare between types of victimization in country of origin, en route in Sinai and in Israel with demographic variables. Chi Square tests were also conducted to assess if trauma victims would be at greater risk for PTSD than non-trauma victims. *t*-test analyses were conducted to assess the differences between trauma victims and non-trauma victims for the means of clinical consultations. Multiple *t*-test analyses were conducted, separately for trauma en route to Israel (Sinai), in Israel and in country of origin, and separately for types of trauma when warranted. Chi square analyses were also conducted to assess the frequencies of trauma victims’ medication, hospitalization, and suicide ideation. Pearson correlations were used to assess relationships between variables. All statistical analyses were deemed significant at a *p* < 0.05 level. We built an index summing the locations where traumatic events were experienced. This Trauma Index ranged between 0 (no trauma) to 3 (trauma en route, trauma in Israel and trauma in country of origin).

SPSS version 23 (IBM, Amonk, NY, USA) was used for all analyses.

## 3. Results

### 3.1. Participants

This study is based on a retrospective chart review, based on medical records of 271 patients who were asylum seekers living in Israel and sought treatment at the *Gesher* Clinic from 2014–2016. The 52 patients who did not report whether they had been victims of trauma were excluded, so 219 asylum seekers (149 men, 68%) were included in the study. Their ages ranged from 20–61 (M = 32.7, SD = 7.65). Most (*n* = 134, 61.2%) were from Eritrea, 56 (25.6%) were from Sudan, 3 were from Ethiopia and 26 did not report their country of origin. Over half (*n* = 126, 57.5%) were unemployed. Of the 78 employed patients, 46 (21.0%) reported working full time. Most asylum seekers (*n* = 141, 64.4%) were Christian, 47 (21.5%) were Muslim and 31 (14.1%) did not report their religion. Participants reported receiving 0–20 years of education (M = 8.83, SD = 3.71) and had 0–9 children (M = 1.09, SD = 1.63), of whom 0–4 (M = 0.49, SD = 0.83) were born in Israel. Most (*n* = 125, 57.1%) were single, 78 (35.6%) were married, 11 divorced, and 3 widowed. Two of the single patients, 32 of those married, and one of those divorced had partners in Israel.

### 3.2. Traumatic Events in Country of Origin

Almost half (42.9%, *n* = 94) the asylum seekers reported having experienced traumatic events in their country of origin. Political persecution was reported by 38 (17.4%), physical or sexual violence by 24 (11.0%), military trauma by seven (3.2%), and civil trauma by 17 (7.8%). Table 1 shows demographic variables across different types of traumatic events in country of origin.

Significant differences were found for gender and country of origin. More men than women had suffered political persecution, military trauma and civil trauma. Asylum seekers from Eritrea suffered more traumatic events than those from Sudan, except for civil war.

### 3.3. Traumatic Events en Route to Israel

There were significant differences for gender and country of origin across asylum seekers who experienced different types of trauma en route to Israel (in the Sinai Peninsula). Only about one quarter of the non-victims were women, and almost two-thirds from Eritrea. Most smuggling, torture and trafficking victims were men, whereas most sexual violence and torture victims were women. Approximately three-quarters of the torture victims and the trafficking and torture victims, and over 80% of the sexual violence victims were from Eritrea. There were no significant between-group differences for family status, religion, employment or partner in Israel (see Table 2).

### 3.4. Traumatic Events in Israel

Only 23 (10.5%) of the asylum seekers reported experiencing traumatic events in Israel. Eleven (~48%) were from Eritrea and 13 (~57%) were Christian. Nine (4.1%) reported having experienced sexual or physical violence in Israel. Fourteen (6.4%) of the asylum seekers reported having experienced civil trauma in Israel. Ten (71.4%) were male, 12 (85.7%) had no partner in Israel, half were married, and 7 (58.3%) were from Eritrea. Half were Muslim and most (*n* = 10, 71.4%) were unemployed.

**Hypothesis** **1** **(H1)**. *Trauma victims would be at greater risk for PTSD than non-trauma victims.*

To assess differences between trauma victims and non-trauma victims for PTSD diagnosis, multiple chi square analyses were conducted, separately for trauma en route to Israel (Sinai), in Israel, and in country of origin, and then separately for types of trauma in each location.

A PTSD diagnosis was given to 96 (43.8%) of the asylum seekers. Trauma was experienced by 114 (52.1%) asylum seekers in Sinai, 23 (10.5%) in Israel and 94 (42.9%) in their country of origin. Asylum seekers who experienced trauma in Sinai were more likely to be diagnosed with PTSD than those who did not, and asylum seekers who experienced trauma in their country of origin were more likely to be diagnosed with PTSD than those who did not (see Figure 1). The number of asylum seekers who reported traumatic events in Israel was insufficient for statistical analysis (e.g., 4.1% [9] were victims of physical/sexual violence, and 6.4% [14] were victims of civil trauma). It was not possible to perform statistical analyses between groups, because the expected cells in chi square were in some cases less than 5.

Fifty-one (23.3%) asylum seekers reported no trauma, one of whom received a PTSD diagnosis. Approximately half (111, 50.7%) experienced trauma at one location, 51 (23.3%) at two locations and 6 (2.7%) at all three places of trauma. The Trauma Index of asylum seekers with a PTSD diagnosis (M = 1.33, SD = 0.54) was significantly higher than that of those without a PTSD diagnosis (M = 0.84, SD = 0.83; *t*(217) = −5.07, *p* < 0.001).

Victims of any kind of torture en route to Israel (torture, trafficking and torture/sexual violence and torture) were more likely to receive a diagnosis of PTSD than those who did not experience torture en route to Israel (*X^2^*(_5_) = 57.89, *p* < 0.001; see Figure 2). The vast majority of smuggling victims had no PTSD diagnosis.

No significant differences were observed in the frequency of PTSD diagnoses across different types of trauma in Israel. Victims of political persecution or physical/sexual trauma in their country of origin were significantly more likely to receive a PTSD diagnosis than non-victims of political persecution or physical/sexual trauma (*X^2^*(_4_) = 15.07, *p* = 0.005; see Figure 3).

**Hypothesis** **2** **(H2).**
*Trauma victims would have more clinical consultations than non-trauma victims.*


The number of clinical consultations ranged from 1–45 (M = 9.29, SD = 9.21). To assess differences between trauma victims and non-trauma victims for clinical consultations, multiple t-test analyses were conducted, separately for trauma en route to Israel (Sinai), in Israel and in country of origin, and separately for types of trauma when warranted.

There was no significant difference in the number of clinical consultations between asylum seekers who experienced trauma en route to Israel and those who did not, or between those who experienced trauma in Israel and those who did not. However, the 94 asylum seekers who experienced trauma in their country of origin had significantly more clinical consultations (M = 11.82, SD = 10.27) than the 125 who did not (M = 0.49, SD = 0.73, *p* < 0.001). Asylum seekers who suffered civil trauma in their country of origin had the highest number of consultations (M = 13.27, SD = 12.53), followed by victims of physical/sexual trauma (M = 11.29, SD = 8.96), political persecution (M = 11.00, SD = 10.15), military trauma (M = 9.14, SD = 8.21), and non-trauma victims (M = 7.49, SD = 7.95; F (4193) = 2.52, *p* = 0.04). There was a positive, significant association between the Trauma Index and number of clinical consultations (r = 0.25, *p* < 0.001).

**Hypothesis** **3** **(H3).**
*Trauma victims would be prescribed more psychiatric medication than non-trauma victims.*


Most participants (*n* = 190, 86.8%) were prescribed psychiatric medication. To compare trauma victims with non-trauma victims, multiple chi square analyses were conducted. No significant between-group differences emerged for different locations or types of traumas. However, participants who were prescribed psychiatric medication had a higher number of clinic consultations (M = 9.82, SD = 9.19) than those who did not (M = 6.52, SD = 9.11, *t*(200) = −1.68, *p* = 0.045).

**Hypothesis** **4** **(H4).**
*Trauma victims would report more drug and alcohol use than non-trauma victims.*


Only 18 (8.2%) of the asylum seekers reported using alcohol and 10 (4.6%) reported using drugs. To assess differences between trauma victims and non-trauma victims for alcohol and drug use, multiple chi square analyses were conducted. No between-group differences for alcohol or drug use were observed for type or place of trauma or for the Trauma Index.

**Hypothesis** **5** **(H5).**
*Trauma victims would report more suicide ideation than non-trauma victims.*


Only 15 (6.8%) of the asylum seekers reported suicidal ideation. To assess differences between trauma victims and non-trauma victims for suicidal ideation, multiple chi square analyses were conducted, for place of trauma and then for types of trauma. No significant between-group differences were found for trauma en route to Israel or in country of origin, and no significant association was found between suicide ideation and the Trauma Index. However, significantly more asylum seekers who suffered trauma in Israel experienced suicidal ideation (*n* = 7, 33.3%) than those who did not (*n* = 8, 4.7%; *X^2^*(_1_) = 21.48, *p* < 0.001). As can be seen in Figure 4, fully 75% of asylum seekers who suffered physical or sexual violence in Israel reported suicidal ideation.

**Hypothesis** **6** **(H6).**
*Trauma victims would have more psychiatric hospitalizations than non-trauma victims.*


Twenty-seven (12.3%) of the asylum seekers had been hospitalized in psychiatric units in Israel. To assess differences between trauma victims and non-trauma victims for psychiatric hospitalization, multiple chi square analyses were conducted, separately for place of trauma and then for type of trauma. No significant difference was found for trauma in Israel or in country of origin. However, significantly fewer asylum seekers who had suffered trauma en route to Israel (*n* = 6, 5.5%) reported a history of psychiatric hospitalizations than those who had not (*n* = 21, 21.4%; *X^2^*(_1_) = 11.54, *p* < 0.001). This finding was in the opposite direction to our hypothesis.

**Hypothesis** **7** **(H7).**
*Emotional support (i.e., having a partner in Israel or being employed) would be associated with fewer PTSD diagnoses, clinic consultations, psychiatric medication prescriptions, drug and alcohol use, suicidal ideation, and psychiatric hospitalizations.*


Chi-square analyses were conducted to assess whether having a partner in Israel (yes/no) and/or being employed (yes/no) were related to PTSD diagnoses, number of consultations, psychiatric medication, drug and alcohol use, suicidal ideation and/or psychiatric hospitalizations. No significant differences were found between having a partner in Israel and the other variables. A significant association was observed between employment status and alcohol use. More currently employed participants (*n* = 11, 14.7%) abused alcohol than currently unemployed participants (*n* = 7, 5.7%; *X^2^*(_1_) = 4.52, *p* = 0.03). This finding was in the opposite direction to our hypothesis.

## 4. Discussion

This study aimed to examine the psychiatric status and mental health care of asylum seekers in Israel, mostly from Eritrea and Sudan, who received psychiatric treatment at the *Gesher* Clinic in Israel between 2014 and 2016. We compared the profiles of those who had experienced traumatic events with those who had not, since this distinction has policy implications in Israel.

Only asylum seekers whom the State of Israel recognizes as victims of trafficking and slavery are entitled to medical care and rehabilitation services, yet even these are limited in duration [21] The most surprising finding was, perhaps, that in contrast to our hypotheses, there were few significant differences in the psychiatric characteristics that differentiated between trauma victims and non-trauma victims, and between asylum seekers with and without PTSD. Those with PTSD were no more likely than those without PTSD to take psychiatric medications, use psychoactive drugs and alcohol, have suicide ideation, and report a history of psychiatric hospitalizations. All asylum seekers included in this study were in mental distress, whether or not they reported having experienced a traumatic event or were diagnosed with PTSD. All had turned to a psychiatric clinic for help, and it, therefore, seems arbitrary to restrict access to psychiatric care to the few whom the State of Israel has recognized as refugees, or victims of human trafficking or slavery.

The high prevalence of PTSD among trauma victims is in line with previous studies highlighting the vulnerability of asylum seekers and refugees to cumulative trauma [25,31,44,45]. Traumatic experiences occur in asylum seekers’ countries of origin, along migratory routes, and in their post-migratory environment.

Among asylum seekers who were victim of trauma, the highest prevalence of PTSD was observed among those who experienced trauma in the Sinai Peninsula en route to Israel. This supports previous findings that asylum seekers who crossed the Sinai desert were victims of a wide range of traumas [6]. They were found to be at high risk for mental health problems [46], and prolonged exposure to traumatic events greatly increased their risk of developing PTSD [47]. Similar risk was found in asylum seekers from various countries in Switzerland, who had experienced or witnessed torture [48]. The low rate of PTSD observed among victims of smuggling may stem from the complicated relationship between smuggling and trafficking—some incidents of smuggling turn into trafficking and some incidents of trafficking turn into smuggling [10]. It may be that being smuggled may scar the souls of asylum seekers less than previous or subsequent traumatic experiences.

Political persecution and physical/sexual trauma in the asylum seekers’ country of origin were risk factors for PTSD. Cumulative documentation from victims of political persecution and physical or sexual violence in Eritrea and Sudan corroborates the depth of their vulnerability from these forms of trauma [49,50,51]. Asylum seekers who had experienced trauma in their country of origin also had more clinical consultations than those who did not, underscoring the heavy residue of trauma in their country of origin.

No specific trauma in the host country, Israel, was found to be associated with PTSD. This may be due to the small number of asylum seekers who reported experiencing trauma in Israel. In addition, many asylum seekers may experience high distress due to post-migration difficulties, whether or not they experienced traumatic events on their migratory path [6,37,52]. Such post-migration difficulties include socioeconomic, social, and interpersonal factors, asylum-related bureaucracy, immigration policy [53], and perceived threat of detention and deportation [54]. Future research should examine the relative impact of specific postmigration stressors on PTSD and mental health.

Asylum seekers who were victims of trauma and diagnosed with PTSD had experienced traumas in more places than those without PTSD. Traumatic events were most experienced in the Sinai Peninsula, followed by country of origin. This supports previous findings that experiencing trauma in more than one location increases risk for PTSD and other disorders among asylum seekers [55,56]. Experiencing a traumatic event in the Sinai desert conferred risk for PTSD, underscoring the severe and prolonged injuries suffered by the asylum seekers in Sinai [6,47]. However, experiencing an additional traumatic event elsewhere along the asylum seekers’ journey did not increase this risk further. This suggests that the asylum seekers’ overall subjective experience of having been traumatized, regardless of the number of their traumatic experiences, confers risk for developing PTSD [57].

Whereas both men and women commonly experience torture, trafficking and sexual violence during their migratory journeys, particularly in Africa [19,58], certain traumas appear to be gender-based. Most torture, trafficking and smuggling victims in our study were men, whereas most sexual violence and torture victims were women. Similar findings were reported in other studies on this population [6,59], and asylum seekers in other Western countries [60,61,62].

Fully 43 (8%) of the trauma victims in our study received a PTSD diagnosis, underscoring the devastating effects of trauma such as torture, trafficking, and physical and sexual violence [6,28]. However, 37% did not. Whereas resilience is a possible explanation, it is important to keep in mind that all participants were in treatment for mental problems at the *Gesher* Clinic. One non-trauma victim was diagnosed with PTSD. She may have omitted to report a traumatic event or been misdiagnosed, possibly due to cross-cultural variation in the presentation of posttraumatic symptoms [63]. Similarly, at least some of the non-trauma victims may not have reported traumatic events they experienced.

In contrast to our hypothesis, a similar percentage of trauma and non-trauma victims were prescribed psychiatric medications. This was no doubt because they were prescribed to the vast majority of asylum seekers. Medications may have been needed, available, convenient, and over-prescribed. We unfortunately had no access to reliable information on the patients referred to individual or group psychotherapy; however, these may be under-used and regarded with mistrust because of differences in cultural attitudes [36,64].

Number of clinical appointments does not necessarily indicate a need for treatment or a level of distress [64]. Prescriptions for psychiatric medications require monitoring, which may explain the association between the number of clinical consultations and the prescription of psychiatric medications. In Israel, at the time of the study, the movement of asylum seekers was restricted and the renewal of medical prescriptions was a common pretext for freedom of movement [65].

Notably, no differences in alcohol or drug use were found between asylum seekers with and without PTSD, which can perhaps be attributed to low rates of report and self-disclosure. Low rates of substance and alcohol abuse were also reported among Russian, Somali and Kurdish migrants in Finland [66], and refugees from former Yugoslavia and the Middle East in Sweden [67]. Although some studies have observed massive drug use in some ethnic minorities in the US [68], Muslim refugees may be protected against substance use and misuse [29]. In this study, employed asylum seekers reported using more alcohol than the unemployed, possibly because they could pay for them.

Suicide ideation was not reported more frequently by asylum seekers who were victims of trauma and those who were non-trauma victims, whether in the country of origin or en route to Israel. Suicide ideation may have been underreported in our study, since there are high rates of suicidality among asylum seekers in Australia [23] and the U.S. [69], among moderate and severe levels of distress among Afghanistan and Syrian asylum seekers in Sweden [70], and detention can increase the risk for suicide ideation in asylum seekers [71]. Yet, most (75%) of victims of physical/sexual violence in Israel reported suicidal ideation. Post-resettlement violence was found to have a larger association with mental health symptoms than pre-resettlement exposure, among Somali refugees in the US and Canada [72]. Trauma-affected refugees in Germany who experienced an additional stressful life event during treatment had more severe symptoms of PTSD, anxiety, and depression than those without a renewed event [73]. Experiencing a new trauma, especially physical or sexual violence in the “land of refuge”, a place that had offered hope for a better future, may lead to despair to the point of wanting to die. Whereas physical/sexual violence experienced in one’s country of origin or en route is traumatic, it may still leave hope for a better future.

Asylum seekers with and without PTSD had similar numbers of psychiatric hospitalizations in Israel. Surprisingly, victims of trauma in Sinai with PTSD reported fewer hospitalizations than those without PTSD. This may suggest resilience, although factors such as socio-demographic and ecological variations in their post-migratory environment, possible alternatives to psychiatric admission, like the Open Clinic of Physicians for Human Rights [47], and traditional healers and religious leaders [36,74] may explain this finding.

Previous research found that emotional support from a partner [75,76] and employment [77,78] in the host country was protective against PTSD. However, we did not find having a partner in Israel or being employed to be associated with lower risk for PTSD, psychiatric medications, clinical consultations, alcohol abuse, suicide ideation, or psychiatric inpatient admissions. A study examining the association between perceived social support and posttraumatic symptoms among Eritrean and Sudanese male asylum seekers in Israel found a significant negative link between the two, but only for men who reported low exposure to traumatic events [36]. In our study, 76.7% of the asylum seekers reported experiencing at least one traumatic event at some point along their process of migration, which may explain why having a partner was not connected to lower distress levels. It is known that the levels of torture experience in the Sinai Desert by trauma victims from Eritrea and Sudan are extreme [10,59].

The lack of association between PTSD diagnosis and employment may be explained by the fact that over half the participants (57.5%) were unemployed and almost 80% of the employed worked part-time, presumably not out of choice. Economic opportunities such as the right to work and access to full employment have been found to have a linear relationship with mental health among refugees and asylum seekers [78]. However, earnings from part-time employment in Israel are not high and does not generally provide economic security in Israel. Part-time employment may even render asylum seekers in Israel vulnerable to trafficking and abuse. Employment may not have been protective of PTSD in this study because most asylum seekers were employed part-time. Moreover, a residence permit is a precondition for asylum seekers to obtain a work permit in Israel. Residence permits are temporary and must be periodically renewed at the Ministry of the Interior [21], which adds to the burden and stress of employed asylum seekers.

This study has several limitations. First, given the cross-sectional nature of the data, causality cannot be determined. Second, the findings are based on a clinical sample of asylum seekers who sought psychiatric help at a specific clinic. It is therefore unclear whether the findings can be generalized to a broader population of asylum seekers. Third, the asylum seekers were assessed by different mental health workers and the inter-judge reliability of diagnoses and other information recorded in the files was not assessed. Professionals based their assessments on patients’ self-report so some information may have been biased or omitted.

## 5. Conclusions

Although this is a clinical sample, with a relatively small number of participants, the findings of this study add to our knowledge about the psychiatric status of asylum seekers seeking mental health care in Israel. The study examined differences between trauma victims and non-trauma victims, and between trauma victims with and without PTSD. Although trauma victims received more PTSD diagnoses than non-victims, no differences were observed between the psychiatric status and mental health care of victims versus non-victims of trauma among asylum seekers in Israel from East Africa. Asylum seekers are a vulnerable population, and it is vital to understand that the risks and hazards they face before, during, and after their migration journey increase the risk for physical and mental problems among trauma victims and non-trauma victims alike. It may be of value to integrate cultural and societal-structural approaches such as the Socio-Cultural Formulation [79] into the psychiatric assessment of asylum seekers. This could help make it clear that they are vulnerable and deserving of psychiatric treatment and psychological support. In any case, clinicians should examine the mental health status of asylum seekers and victims of human trafficking with thoroughness and dignity, and ensure that they receive the help warranted by their experiences and needs.

From a clinical perspective, the lack of difference between the psychiatric status and mental health care of victims versus non-victims in this study strongly support a recent initiative of the Israel Ministry of Health [80]. According to this initiative, the Israeli government would provide health insurance for asylum seekers [80] and abandon the narrow interpretation of the Refugee Convention [3] adopted to date in the field of health. The results of our study strengthen the case for establishing organized, coordinated, sustainable mental health services that serve the needs of the community of asylum seekers in Israel. It would seem to be appropriate for the Ministry of Health or a Health Maintenance Organization to extend tailored services to meet the needs of the community.

The few clinical differences we observed between asylum seekers who were victims of human trafficking, torture, and other atrocities and asylum seekers who were non-trauma victims challenges the narrow interpretation currently given by the State of Israel to the international conventions it ratified. The Refugee Convention [3] and the Palermo Protocol [41], for example, enable Israel to recognize only 0.06% of all asylum seekers’ applications for refugee status [80]. This stance joins local and international voices calling for a review of the Palermo Protocol [41] and the definition of human trafficking [81,82,83], so as to “… protect and assist the victims of such trafficking, with full respect for their human rights” ([41]; Article 2(b)), and help asylum seekers gain access to appropriate services, regardless of their visa status.

## Figures and Tables

**Figure 1 ijerph-18-10661-f001:**
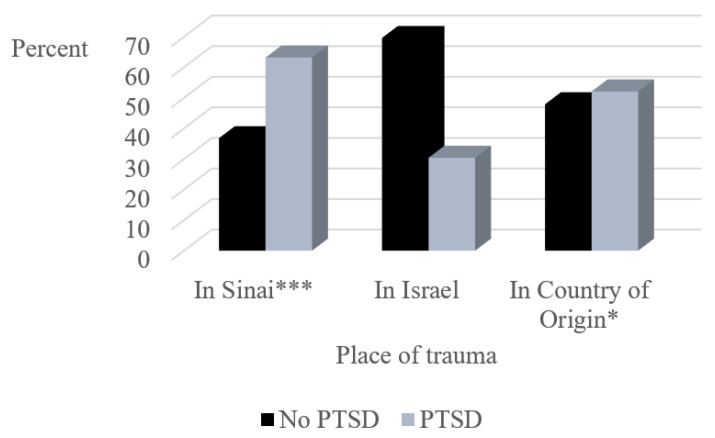
Percentage of asylum seekers with PTSD across place of trauma. *** Significant at *p* < 0.001, * Significant at *p* < 0.05. PTSD = Post-traumatic stress disorder.

**Figure 2 ijerph-18-10661-f002:**
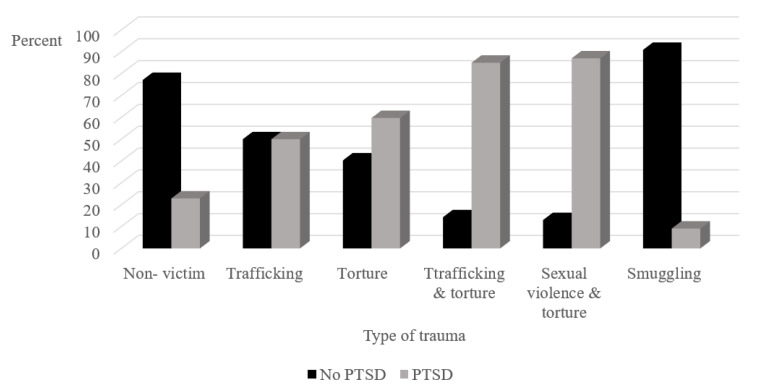
Percentage of asylum seekers who experienced trauma en route to Israel, with and without PTSD as a function type of trauma. PTSD = Post-traumatic stress disorder.

**Figure 3 ijerph-18-10661-f003:**
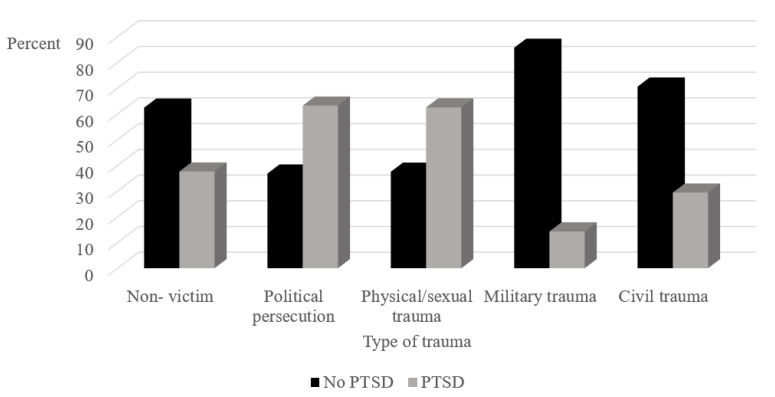
Percentage of asylum seekers with and without PTSD across different types of trauma in country of origin. PTSD = Post-traumatic stress disorder.

**Figure 4 ijerph-18-10661-f004:**
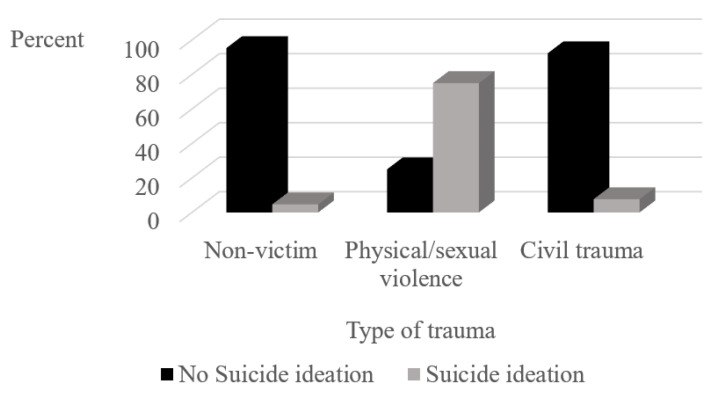
Percentage of asylum seekers with and without suicide ideation as a function type of trauma in Israel (*n* = 193).

**Table 1 ijerph-18-10661-t001:** Demographic variables across different types of traumatic events in country of origin.

Type of Victim	Gender: Men *n* (%)	Spouse in Israel *n* (%)	Family Status *n* (%)			Country of Origin *n* (%)			Religion *n* (%)	Employed (%)
			Single	Married	Other	Ethiopia	Eritrea	Sudan	Muslim	Yes
No traumatic events *n* = 125	80 (64.5)	24 (20.2)	72 (57.6)	45 (36.0)	8 (6.4)	1 (0.9)	83 (76.9)	24 (22.2)	23 (21.5)	45 (38.1)
Political persecution *n* = 38	32 (88.9)	3 (8.8)	22 (59.5)	13 (35.1)	2 (5.4)	0 (0)	24 (68.6)	11 (31.4)	8 (25.0)	9 (25.0)
Physical/sexual violence *n* = 24	12 (52.2)	4 (18.2)	12 (52.2)	9 (39.1)	2 (8.7)	1 (5.0)	13 (65.0)	6 (30.0)	3 (15.8)	10 (50.0)
Military trauma *n* = 7	6 (85.7)	0 (0)	5 (71.4)	2 (28.6)	0 (0)	0	6 (100)	0 (0)	1 (16.7)	3 (42.9)
Civil trauma *n* = 17	14 (82.4)	2 (12.5)	11 (64.7)	4 (23.5)	2 (11.8)	(0)	5 (31.3)	11 (68.8)	8 (50.0)	7 (43.8)
Significance	χ^2^_(4)_ = 13.3 *p* = 0.01	NS	NS			χ^2^_(8)_ = 21.00 *p* = 0.007			NS	NS

**Table 2 ijerph-18-10661-t002:** Demographic variables of asylum seekers who experienced different types of trauma in the Sinai Peninsula, and non-trauma victims.

Table	Gender: Men *n* (%)	Spouse in Israel *n* (%)	Family Status *n* (%)			Country of Origin ** *n* (%)			Religion *n* (%)	Employed *n* (%)
			Single	Married	Other	Ethiopia	Eritrea	Sudan	Muslim	Yes
Non-victims * *n* = 105	74 (71.8)	82 (84.5)	57 (54.8)	39 (37.5)	8 (7.7)	1 * (1.2)	53 (63.1)	30 (35.7)	25 (30.8)	42 (43.8)
Trafficking *n* = 4	0	2 (50.0)	2 (50.0)	1 (25.0)	1 (25.0)	1 (50.0)	1 (50.0)	0 (0)	0	1 (25.0)
Torture *n* = 62	50 (83.3)	48 (84.2)	38 (62.3)	20 (32.8)	3 (4.9)	0 (0)	45 (75.0)	15 (25.0)	12 (22.2)	19 (33.3)
Trafficking & torture *n* = 14	9 (64.3)	12 (92.3)	11 (78.6)	3 (21.4)	0 (0)	0 (0)	10 (76.9)	3 (23.1)	3 (27.3)	5 (38.5)
Sexual violence & torture *n* = 23	7 (30.4)	16 (69.6)	10 (43.5)	11 (47.8)	1 (4.3)	1 (4.3)	19 (82.6)	3 (13.0)	0 (0)	9 (39.1)
Smuggling *n* = 11	9 (81.8)	10 (90.9)	7 (63.6)	4 (36.4)	0 (0)	0 (0)	6 (54.5)	5 (45.5)	4 (40.0)	2 (18.2)
Significance	χ^2^_(5)_ = 32.2 *p* <0.001	NS	NS			χ^2^_(10)_ = 40.04 *p* < 0.001			NS	NS

* Only 84 non-victims reported their country of origin. ** Included in the table are only participants for whom we had data concerning country of origin and type of victimization.

## Data Availability

The data is available on demand from the corresponding author.

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
