# Peer review of "Trauma, Post-Traumatic Stress Disorder, and Mental Health Care of Asylum Seekers"

_ijerph, 2021, doi:10.3390/ijerph182010661_

Round 1

Reviewer 1 Report

This manuscript highlights the mental health experiences and needs of asylum seekers in Israel.  It is a solid manuscript and provide an interesting analysis of data from a retrospective chart review.  The information offers a clear conceptualization about how asylum seekers may present differently than refugees and trafficked individuals in their mental health needs.  That said, it is not clear why this information and the findings are new or novel.  Is it because the study was done in the Israeli context?  If so, I would recommend providing more information about what could be offered to asylum seekers from a healthcare and/or clinical settings to support their move to more positive mental health. How would clinicians address these issues for their clients? Was there any information in the charts about interventions for these asylum seekers? It might be helpful to tease out what type of trafficking individuals experienced, e.g., sexual, forced labor, etc. On page 10, the statement that "there were few significant differences between trauma victims and non-trauma victims, and between asylum seekers with and without PTSD" is concerning. Could this be due to a lack of exact information in the records about the type of trauma or is it that the analysis did not considering the specifics of different trauma types? On page 13 prior to the Conclusion section, there is a discussion about "the lack of association between PTSD diagnosis and employment." Why is this lack of association "explained" by part-time work? Please clarify.

Reviewer 2 Report

Overall summary

The authors conducted a study to examine the psychiatric status of asylum seekers, mostly from Eritrea and Sudan, who arrived in Israel through the Sinai Peninsula between 2007 and 2013.  Furthermore, they characterized the types of traumas the asylum seekers experienced and the mental health care they received at a psychiatric clinic established by the Israeli Ministry of Health in 2014.

General comments

The article is interesting and has great clinical, economic, and social relevance. However, there are deficiencies which should be addressed before publication could be considered.

  1. Introduction
    1. This section is too extensive, so from my point of view it should be rewritten. Authors must remember that introductions are not exhaustive reviews of the extant literature surrounding the topic of study. In this case, the authors include too much information about migrants which is not necessary to understand the manuscript. The introduction should focus on asylum seekers in Israel.
    2. Line 41. Review the reference #4: Data about the number of refugees and of asylum seekers worldwide has been updated.

  1. Materials and methods
    1. Please, clarify if the study has included the entire population possible (n= 271-52= 219). Two hundred and nineteen participants are not too much compared to the number of asylum seekers who arrived in Israel through the Sinai Peninsula from Sudan and Eritrea.
    2. Lines 153-161. This information must be included in the section of results, instead of in the section of “materials and methods”.
    3. The “Subsection 2.4. Data analysis” must be completed: a) how the results are going to be presented (mean, percentages…); b) the level of significance to be used is missing; c) Lines 205-206: T-test analyses were conducted not only to assess the differences between trauma victims and non-trauma victims for clinical consultations (see lines 266-267 and lines 307-308), so this information also must be included here; d) besides chi square and T-test analyses, other tests were used (ie, correlation; see lines 299-300); e) the explanation about how to calculate the Trauma Index must be included here, instead of in the section of results (lines 260-262).

  1. Results
    1. Tables 1 and Table 2: Taking into account the data registered for Ethiopia, I cannot understand how the authors could study the differences for country of origin. In general, the “n” is <5 what is a problem for chi-square tests. The same for data included in hypothesis 1d (total n=18, subgroups??) and 1e (total n=15, subgroups??)
    2. Subsection 3.3. Traumatic events in Israel: In contrast to the subsections 3.1 and 3.2, the authors do not include information about potential differences between groups. In this subsection, significant differences were not studied between groups?
    3. Lines 253-256 and figure 1. Can you explain the differences found for asylum seekers who experienced trauma in Israel. Only information for asylum seekers who experienced trauma in Sinai or in their country of origin was included.
    4. All the figures (including the figure legends) must be improved. Ex. Abbreviations and symbols must be explained (figures 1-3), some words are cut off (figures 2 and 3), it is not clear how many people are represented by each bar …"

  1. Discussion
    1. I do not know if it is a misunderstanding but taking into account the results of the hypothesis 1a, I cannot understand why the authors conclude “there were few significant differences between trauma victims and non-trauma victims” (lines 360-361). And in fact, in lines 370-371 authors say: “the high prevalence of PTSD among trauma victims is in line…”
    2. Taking into account that “some incidents of smuggling turn into trafficking and some incidents of trafficking turn into smuggling” (lines 382-383), the authors could have included only one category (trafficking & smuggling) instead two ones (trafficking, smuggling). Would the results have been more reliable?
    3. Please, review the lines 442-443 because here the authors say that “employed asylum seekers reported using more alcohol and drugs than the unemployed”; however, according to the results included in the lines 349-350, a significant association was only observed between employment status and alcohol use (you do not mention “drug use”).

Other comments

  1. Lines 270, 278, 324, 339, 351: Typo errors
  2. Line 340: Figure 5 is missing
  3. Line 2: All the abbreviations must be explained in the text the first time they are used, included the title.
  4. Please, include a reference after the sentence “only asylum seekers….services (lines 358-359).

Round 2

Reviewer 2 Report

The manuscript has been improved but the authors have not addressed all the changes which were previously suggested.

Below, I have included the comments that should be reviewed.

  1. Introduction

This section is too extensive, so from my point of view it should be rewritten. Authors must remember that introductions are not exhaustive reviews of the extant literature surrounding the topic of study. In this case, the authors include too much information about migrants which is not necessary to understand the manuscript. The introduction should focus on asylum seekers in Israel.

The introduction has been shortened and now focuses on asylum seekers in Israel.

The introduction has been barely modified, so it is still vague. Deleting some sentences, it is not enough. This comment should be reviewed.

  1. Results
    1. Subsection 3.3. Traumatic events in Israel: In contrast to the subsections 3.1 and 3.2, the authors do not include information about potential differences between groups. In this subsection, significant differences were not studied between groups?

Since there were only 23 asylum seekers who reported traumatic events in Israel, it was not possible to perform statistical analyses between groups. The expected cells in Chi-square were in some cases less than 5.

This must be clarified in the manuscript.

  1. All the figures (including the figure legends) must be improved. Ex. Abbreviations and symbols must be explained (figures 1-3), some words are cut off (figures 2 and 3), it is not clear how many people are represented by each bar …"

The Figures have been improved in line with this comment. 

In figure legends the word “Abbreviations” it is better than “note”. On the other hand, you did not clarify what * or *** means (figure 1), and some words are still cut off (eg. Sexual violence &…) (figures 2 and 3).

This comment must be reviewed.

Author Response

Reviewer 2:

Introduction

The introduction has been barely modified, so it is still vague. Deleting some sentences, it is not enough. This comment should be reviewed.

The introduction has been further shortened and modified.

Results

  1. It must be clarified in the manuscript that “Since there were only 23 asylum seekers who reported traumatic events in Israel, it was not possible to perform statistical analyses between groups. The expected cells in Chi-square were in some cases less than 5.”

This has been added to the text (p. 6).

  1. In figure legends the word “Abbreviations” it is better than “note”. On the other hand, you did not clarify what * or *** means (figure 1), and some words are still cut off (eg. Sexual violence &…) (figures 2 and 3).

This comment must be reviewed.

“Note” has been replaced by “Abbreviations” in the figure legends. The meaning of * and *** has been added to Figure 1. All words are now visible in Figures 2 and 3.
